# HyperIV: Real-time Implied Volatility Smoothing

**Yongxin Yang** [1]  **Wenqi Chen** [1]  **Chao Shu** [1]  **Timothy Hospedales** [2]

## Abstract

We propose HyperIV, a novel approach for real-time implied volatility smoothing that eliminates the need for traditional calibration procedures. Our method employs a hypernetwork to generate parameters for a compact neural network that constructs complete volatility surfaces within 2 milliseconds, using only 9 market observations. Moreover, the generated surfaces are guaranteed to be free of static arbitrage. Extensive experiments across 8 index options demonstrate that HyperIV achieves superior accuracy compared to existing methods while maintaining computational efficiency. The model also exhibits strong cross-asset generalization capabilities, indicating broader applicability across different market instruments. These key features – rapid adaptation to market conditions, guaranteed absence of arbitrage, and minimal data requirements – make HyperIV particularly valuable for real-time trading applications. We make code available at https://github.com/qmfin/hyperiv.

## 1. Introduction

The implied volatility surface is a fundamental concept in options pricing. Market participants prefer to quote prices in terms of implied volatilities, as these values are more stable over time and provide an exact transformation to option prices. These implied volatilities serve as crucial inputs for more sophisticated models, which are then used to value exotic derivatives, determine margin requirements, and provide liquidity in options markets. At its core, constructing an implied volatility surface involves converting discrete option price quotes into a continuous, smooth surface that spans all possible combinations of strike prices and maturities. This process is commonly known as implied volatility smoothing. The construction faces several key challenges:

(i) the surface must be arbitrage-free, thus simply interpolating or extrapolating from potentially arbitrageable quotes does not work; (ii) the model requires frequent recalibration to reflect market movements, requiring computational efficiency; (iii) at higher frequencies, the limited availability of high-quality option prices is often the bottleneck, especially for fitting the tails of volatility curves. These practical constraints have historically favoured simple parametric models (Gatheral, 2004; Gatheral & Jacquier, 2014; Hendriks & Martini, 2019; Mingone, 2022; Zaugg et al., 2024) over machine learning approaches (Ackerer et al., 2020; Zheng et al., 2021; Bergeron et al., 2021; Ning et al., 2023; Gonon et al., 2024).

In this paper, we introduce a more challenging and practically demanding problem: constructing implied volatility surfaces using extremely sparse observations (fewer than 10 contracts) at high frequency (e.g., one-minute interval). This setting better reflects real-world trading conditions, where only a small set of options is actively traded with reliable prices at any given moment, and where surfaces must be generated within milliseconds for time-sensitive applications. To address this challenge, we propose HyperIV, a hypernetwork-based method that constructs implied volatility surfaces for new market conditions in 2 milliseconds, after training on historical data. We validate HyperIV through extensive experiments on both one-minute interval and end-of-day data across diverse assets, including major U.S. indices (S&P 500, NASDAQ 100) and international indices (MSCI World, MSCI Emerging Markets). Our evaluation encompasses over 150,000 surfaces and 50 million data points, demonstrating the model's robustness and generalization capabilities. The results show that HyperIV achieves superior accuracy compared to existing methods while maintaining computational efficiency and arbitrage-free properties.

### 1.1. Literature Review

Implied volatility smoothing has been extensively studied, with approaches generally falling into two categories (Homescu, 2011). The first category comprises indirect methods that initially fit a model for option prices, typically driven by underlying asset dynamics such as (local) stochastic volatility models (Heston, 1993; Hagan et al., 2002), Levy processes (Madan et al., 1998; Kou, 2002; Carr et al., 2002),

---

[1]Queen Mary University of London [2]University of Edinburgh. Correspondence to: Yongxin Yang <yongxin.yang@qmul.ac.uk>.

*Proceedings of the 42nd International Conference on Machine Learning*, Vancouver, Canada. PMLR 267, 2025. Copyright 2025 by the author(s).

or rough volatility models (Christian Bayer & Gatheral, 2016; El Euch & Rosenbaum, 2018; Gatheral et al., 2018). These methods then convert the fitted price surface to an implied volatility surface through the inverse Black-Scholes formula, using root-finding algorithms. The second category consists of direct methods that employ specific closed-form representations, either through carefully designed functions (Gatheral, 2004; Gatheral & Jacquier, 2014; Hendriks & Martini, 2019; Mingone, 2022; Zaugg et al., 2024) or universal function approximators like neural networks (Ackerer et al., 2020; Zheng et al., 2021; Bergeron et al., 2021; Ning et al., 2023; Gonon et al., 2024). Direct methods have gained greater adoption due to their easier calibration and faster execution. For simpler models, such as the Stochastic Volatility Inspired (SVI) approach (Gatheral, 2004) and its extensions (Gatheral & Jacquier, 2014; Guo et al., 2016; Hendriks & Martini, 2019; Martini & Mingone, 2022), ensuring absence of arbitrage can be achieved through constraints on the parameter space (Fukasawa, 2010; Homescu, 2011; Cohort et al., 2019). However, these methods may fail when market conditions fall outside the scope of their chosen functions. For instance, very short-term options markets can exhibit W-shaped volatility curves (Glasserman & Pirjol, 2023), which some models struggle to capture. Neural network-based methods offer greater flexibility in addressing such challenges. Ackerer et al. (2020) proposed a hybrid approach where a neural network *corrects* simpler methods (e.g., SVI). However, enforcing no-arbitrage conditions on neural networks is more complex, typically requiring careful architecture design, auxiliary loss functions, and data augmentation techniques (Zheng et al., 2021). Additionally, these machine learning methods generally assume dense observations and require longer calibration time compared to simpler models. For constructing complete implied volatility surfaces from extremely sparse data, the Longitude algorithm (Zetocha, 2022) addresses strike interpolation and extrapolation for individual maturities by employing the Neri-Schneider entropy maximisation algorithm (Neri & Schneider, 2012), and then handles maturity extrapolation through time translation of log-normal cumulative distribution functions. This concept of transforming distributions across maturities was further developed in (Zetocha, 2023) using an optimal transport-inspired framework for generating arbitrage-free transformations of an implied volatility surface. Building on these approaches, (Cao et al., 2024) further investigated the theoretical foundations, particularly the 'One-X property' and its role in establishing conditions to eliminate calendar arbitrage. They also explored the use of Log Gaussian Mixtures (as opposed to simpler Log Gaussian distributions) for better capturing complex volatility features like W-shapes.

Implied volatility models contain free parameters that require calibration before deployment. This calibration proce-

dure mirrors model training in machine learning, typically involving iterative optimisation from randomly initialised parameters. Deep learning approaches have emerged to accelerate this calibration procedure. Hernandez (2016) pioneered a direct mapping from prices/implied volatilities on a rectangular mesh grid of moneyness and maturity to model parameters. In an inverse approach, Liu et al. (2019); Benth et al. (2021); Blanka Horvath & Tomas (2021) map model parameters to prices/implied volatilities, necessitating a lightweight second-stage optimisation on the neural network inputs (equivalently, model parameters). A key limitation of these approaches is their reliance on fixed moneyness and maturity grids, which rarely align with actual market observations. More recent work by Bayer et al. (2019); Yang & Hospedales (2023); Baschetti et al. (2024) addresses this by adopting a point-wise approach that accepts continuous inputs for moneyness and maturity rather than using fixed grid positions. Our method builds upon these calibration acceleration techniques, eliminating the need for repeated calibration after the initial model training. While similar to (Hernandez, 2016), our approach differs in two key aspects: first, we employ a hypernetwork architecture (Ha et al., 2016) where one neural network generates parameters for a more compact network that models implied volatility surface more accurately and flexibly; second, we handle a variable-sized set of option contracts with continuous moneyness and maturity values, rather than requiring fixed-grid data. Alternative calibration-free methods have been proposed by Bergeron et al. (2021) and Gonon et al. (2024). However, these approaches have not been validated in sparse data scenarios, and adapting them to this new setting needs non-trivial modifications, as we discuss in Section 4.2.

Our technical approach leverages hypernetworks (Ha et al., 2016), which utilise one neural network to generate the parameters of another. This architecture has demonstrated remarkable success across various domains, particularly in implicit neural representations (Sitzmann et al., 2020) where networks learn continuous mappings from coordinates to signals. Recent work by Mundinger et al. (2024) showcases hypernetworks' capability in representing PDE solution operators, suggesting their potential for financial applications. The connection between implied volatility smoothing and implicit neural representations emerges naturally: both domains involve mapping from low-dimensional coordinates (moneyness-maturity pairs or spatial coordinates) to function values (volatilities or signal intensities).

### 1.2. Contributions

We make three main contributions. First, we introduce a new problem setting that echoes few-shot learning in machine learning, constructing complete implied volatility surfaces from fewer than 10 data points. Second, we develop a

hypernetwork-based method that generates arbitrage-free implied volatility surfaces within 2 milliseconds. Third, we perform comprehensive experiments on over 150,000 surfaces and 50 million contracts, demonstrating our method's superior performance compared to several baselines adapted to the same setting.

## 2. Preliminaries

Options are financial contracts that provide the holder with the right, but not the obligation, to buy (call option) or sell (put option) an underlying asset at a specific price (strike price) on a designated future date (maturity date). The underlying assets for options can include indices, stocks, currencies, commodities, and other financial derivatives (e.g., futures or even options themselves). These instruments are primarily used to hedge against price movements of the underlying asset. The two main categories of options are European and American options. European options permit exercise only on the maturity date, while American options allow exercise at any time before maturity. These options are primarily traded on exchanges, such as the Chicago Board Options Exchange (CBOE). There also exist exotic options (e.g., Barrier Options), which are typically traded in the over-the-counter (OTC) market. In this study, we focus on European options with underlying assets consisting of indices, given their widespread market presence.

**Notations** We introduce the notations used throughout this paper. For a given option contract, we use $K$ to denote its strike price, $t$ for its annualised time to maturity, $S$ for the current price of the underlying asset, and $F$ for the forward price of the underlying asset, calculated as $F = (S - D)\mathrm{e}^{rt}$, where $r$ represents the risk-free interest rate and $D$ denotes the dividend. While $r$, $D$ and $F$ are time-dependent, we omit this dependency for notational simplicity. The variables $(K, t, S, r, D)$ are directly observable from the market. However, there exists ambiguity in the option's price due to the presence of both bid and ask prices. In this work, we define the option price $V$ as the mid-point between the best bid and best ask price. Finally, we define the log forward-moneyness as $k = \log(\frac{K}{F})$.

The Black-Scholes model was the first widely used method for pricing European options, characterised by a single parameter – volatility $\sigma \in \mathbb{R}_+$ under the risk-neutral measure. For a European option, the model estimates the theoretical price through the following closed-form formula:

$$V(k, t, F, r, \delta|\sigma) = e^{-rt}F\left(\delta\Phi\left(\delta d_+(k, t|\sigma)\right) - \delta e^k \Phi\left(\delta d_-(k, t|\sigma)\right)\right) \quad (1)$$

where $d_\pm(k, t|\sigma) = \frac{1}{\sigma\sqrt{t}}\left(-k \pm \frac{1}{2}\sigma^2 t\right)$. Here $\delta = 1$ represents a call option, $\delta = -1$ represents a put option, and $\Phi(\cdot)$ denotes the cumulative distribution function of the standard normal distribution. The parame-

ter $\sigma$ is typically calibrated by minimising the squared differences between the model predicted prices and the observed market prices for a finite set of option contracts $\{(k_1, t_1, F_1, r_1, \delta_1, V_1), (k_2, t_2, F_2, r_2, \delta_2, V_2), \dots, (k_N, t_N, F_N, r_N, \delta_N, V_N)\}$, i.e., through the optimisation:

$$\sigma = \operatorname*{argmin}_\sigma \frac{1}{N} \sum_{i=1}^N \left(V(k_i, t_i, F_i, r_i, \delta_i|\sigma) - V_i\right)^2 \quad (2)$$

The assumption of constant $\sigma$ is unlikely to hold in real markets, thus limiting the effectiveness of the Black-Scholes model. However, if we reduce Eq. 2 to a single option contract, we can determine the unique $\sigma$ that matches the option price exactly (up to machine precision):

$$\sigma_i = \operatorname*{argmin}_\sigma \left(V(k_i, t_i, F_i, r_i, \delta_i|\sigma) - V_i\right)^2 \quad (3)$$

The solution exists and is unique because the derivative of $V$ with respect to $\sigma$ (known as Vega in option Greeks) is strictly positive. This $\sigma_i$ serves as a proxy for option price $V_i$ and is referred to as implied volatility. Using $\sigma_i$ instead of $V_i$ offers three key advantages: (i) it eliminates the need to treat put and call options separately, as the implied volatility remains identical for both types with matching strike and maturity; (ii) implied volatility exhibits less dependence on the underlying asset price and provides a more numerically stable range for modelling purposes; (iii) both the shape and level of the implied volatility surface demonstrate remarkable stability, with stylised features such as the volatility smile remaining consistent across different time periods and option contracts.

Implied volatility smoothing refers to the process of constructing a smooth surface that maps pairs of moneyness ($k$) and maturity ($t$) to implied volatility ($\sigma$), using data from a finite number of option contracts. This process typically involves a parametric function $\sigma_\theta$ and requires solving the following optimisation problem:

$$\min_\theta \frac{1}{N} \sum_{i=1}^N \left(\sigma_\theta(k_i, t_i) - \sigma_i\right)^2 \quad (4)$$

In practice, implied volatility errors of the same magnitude can lead to significantly different pricing errors across different moneyness values. To address this, practitioners often use Vega-weighted loss functions. An alternative approach is to minimise errors directly in the price space by transforming the implied volatilities back to option prices:

$$\min_\theta \frac{1}{N} \sum_{i=1}^N \left(V(k_i, t_i, F_i, r_i, \delta_i|\sigma_\theta(k_i, t_i)) - V_i\right)^2 \quad (5)$$

The parametric function $\sigma_\theta(\cdot)$ cannot be arbitrarily specified, as it must meet certain constraints to prevent arbitrage opportunities, which we will examine in detail in Sec. 3.3.

# 3. Methodology

## 3.1. Problem Setting

Eq. 4 is generally not considered a challenging problem under conventional settings for two main reasons: (i) the parameter space of $\sigma_\theta$ is relatively modest, ranging from just a few parameters (Gatheral & Jacquier, 2014) to several hundreds (Zheng et al., 2021) or thousands (Ackerer et al., 2020) – a scale that is considered small in the deep learning era; (ii) most studies use end-of-day (EOD) data for their experiments. For widely traded options, such as the S&P500 index option, EOD data typically contains tens of thousands of contracts, providing sufficient data for model training. Additionally, computational time is less critical since the model needs to be trained/calibrated only once per day.

In this work, we present a more challenging scenario – fitting the implied volatility surface using one-minute interval data with only a small set of observed contracts (typically fewer than 10). This setting better reflects real-world option trading conditions, where within any given minute, only a few hundred contracts might be available, and even fewer will have reliable prices (indicated by reasonably small bid/ask spreads). The model must generate the surface within milliseconds, making traditional neural network training or fine-tuning approaches less practical. Specifically, we propose a model of the following form:

$$\sigma_\theta(k,t) = f_\theta(k,t|\mathcal{Z}) \qquad (6)$$

where $\mathcal{Z} = \{(k_1,t_1,\sigma_1), (k_2,t_2,\sigma_2),\dots\}$ represents a reference set containing fewer than 10 elements. The model parameter $\theta$ can be trained using historical data and remains fixed during deployment.

## 3.2. A Hypernetwork Approach

We propose to model $f_\theta(k,t|\mathcal{Z})$ using a hypernetwork (Ha et al., 2016). Specifically, a hypernetwork, denoted as $g_\theta(\cdot)$, takes a set as input and outputs a flattened vector. This flattened vector is split, and each segment is reshaped appropriately so that a compact network that models the implied volatility surface, $h_\omega(\cdot)$, can utilise them as its parameters. Finally, this compact network produces the estimated implied volatility for any pair of $(k,t)$ within the considered domain. This procedure is represented as follows:

$$\omega = g_\theta(\mathcal{Z}) \quad \sigma = h_\omega(k,t) \qquad (7)$$

Here, $g_\theta(\cdot)$ can be any neural network that handles *sets* (Zaheer et al., 2017), meaning the order of option contracts in $\mathcal{Z}$ does not affect the output. In this work, we choose to use a Transformer encoder (Vaswani et al., 2017) without positional embedding, in the spirit of (Lee et al., 2019), and apply mean pooling over the sample axis. The design requirements for $h_\omega(\cdot)$ are as follows: (i) its output must be

|   | $F$ | $t$ | $r$ | $K$ | $V$ | $k$ | $\sigma$ | $\Delta$ |
|---|------|--------|--------|------|--------|---------|--------|---------|
| 1 | 3827.08 | 0.0192 | 0.0399 | 3755 | 17.00 | -0.0190 | 0.2100 | -0.2519 |
| 2 | 3827.08 | 0.0192 | 0.0399 | 3825 | 42.40 | -0.0005 | 0.2056 | -0.4867 |
| 3 | 3827.08 | 0.0192 | 0.0399 | 3900 | 15.00 | 0.0189 | 0.1955 | 0.2471 |
| 4 | 3836.34 | 0.0822 | 0.0411 | 3675 | 39.45 | -0.0430 | 0.2335 | -0.2498 |
| 5 | 3836.34 | 0.0822 | 0.0411 | 3840 | 92.15 | 0.0010 | 0.2148 | 0.5061 |
| 6 | 3836.34 | 0.0822 | 0.0411 | 4000 | 28.30 | 0.0418 | 0.1936 | 0.2343 |
| 7 | 3853.73 | 0.2384 | 0.0439 | 3575 | 74.40 | -0.0751 | 0.2512 | -0.2503 |
| 8 | 3853.73 | 0.2384 | 0.0439 | 3855 | 163.40 | 0.0003 | 0.2209 | 0.5203 |
| 9 | 3853.73 | 0.2384 | 0.0439 | 4125 | 50.40 | 0.0680 | 0.1917 | 0.2483 |

*Table 1.* A snapshot of the reference set $\mathcal{Z}$

non-negative, achieved by using a softplus activation function in the last layer; (ii) it must satisfy the arbitrage-free constraints, which are detailed in Sec. 3.3. We found that a simple multi-layer perceptron (MLP) with two hidden layers, each containing 16 neurons, performs well after initial experimentation. Notably, this network has only 337 parameters. The architectures of $g_\theta(\cdot)$ and $h_\omega(\cdot)$ are described in detail in Appendix A.

The trainable parameters reside solely within the hypernetwork $g_\theta(\cdot)$, as the parameters for the implied volatility surface network are generated on-the-fly rather than being trained. The objective function is formulated as:

$$\min_\theta \frac{1}{M}\sum_{j=1}^{M}\frac{1}{N^{(j)}}\sum_{i=1}^{N^{(j)}}\left(h_{\omega^{(j)}}(k_i^{(j)},t_i^{(j)}) - \sigma_i^{(j)}\right)^2 \qquad (8)$$

where $\omega^{(j)} = g_\theta(\mathcal{Z}^{(j)})$. In this expression, $j$ iterates over the number of intervals in a historical period, and $i$ iterates over the number of option contracts within a given interval, while $Z^{(j)}$ represents a small set of option contracts for that interval. While the reference set $\mathcal{Z}$ could be constructed through any consistent rules, we select 9 contracts that are closest to $\{$at-the-money, 25 $\Delta$ Call, 25 $\Delta$ Put$\} \times \{$7 days, 1 month, 3 months$\}$ in this work. The rationale behind this selection is that these options correspond to the most liquid contracts in our datasets, ensuring price reliability[1]. A snapshot of the reference set can be found in Table 1.

After training, the model can generate a complete implied volatility surface for any new reference set $\mathcal{Z}$ through a single forward pass of the hypernetwork. This allows real-time implied volatility smoothing for new market conditions without further optimisation.

## 3.3. Arbitrage-free Constraints

The implied volatility surface network $(k,t) \xrightarrow{h_\omega} \sigma$ must comply with arbitrage-free constraints, as discussed earlier. A fundamental requirement in option pricing theory is that the model must be free of static arbitrage – trading strategies

---

[1]Here, $\Delta$ refers to the derivative of the option price with respect to the underlying asset's price. A 25 $\Delta$ implies that a \$1 increase in the underlying asset will lead to an approximate \$0.25 increase in the call option price or \$0.25 decrease in the put option price.

that guarantee a positive profit with non-zero probability while having zero probability of loss (Carr et al., 2003). There are two cases: calendar spread arbitrage and butterfly arbitrage. The absence of calendar spread arbitrage implies the monotonicity of option prices with respect to maturity, while the absence of butterfly arbitrage implies the corresponding density is non-negative.

**Calendar spread arbitrage** We denote the non-discounted call option price from the Black-Scholes model as $\tilde{V} = e^{rt}V = F\left(\Phi(d_+(k,t|\sigma)) - e^k\Phi(d_-(k,t|\sigma))\right)$. An implied volatility surface is free of *calendar spread arbitrage* if, for all $k \in \mathbb{R}$ and $t \in \mathbb{R}_+$, the derivative of $\tilde{V}$ with respect to $t$ is non-negative. Defining $b(k,t) = \frac{\phi(d_-(k,t|\sigma(k,t)))}{\sigma(k,t)\sqrt{t}}$, where $\phi(\cdot)$ represents the probability density function of the standard normal distribution, we obtain:

$$\frac{\partial \tilde{V}}{\partial t} = \frac{K}{2}b(k,t)\sigma(k,t)\left(\sigma(k,t) + 2t\frac{\partial\sigma(k,t)}{\partial t}\right) \quad (9)$$

This leads to the following constraint:

$$\sigma(k,t) + 2t\frac{\partial\sigma(k,t)}{\partial t} \geq 0 \quad \forall k \in \mathbb{R} \text{ and } t \in \mathbb{R}_+ \quad (10)$$

**Butterfly arbitrage** An implied volatility surface is free of *butterfly arbitrage* if the corresponding density is valid, meaning it is non-negative and integrates to one for all time slices:

$$p(k,t) \geq 0 \quad \forall k \in \mathbb{R} \text{ and } t \in \mathbb{R}_+ \quad (11)$$

$$\int_{-\infty}^{+\infty} p(k,t)\mathrm{d}k = 1 \quad \forall t \in \mathbb{R}_+ \quad (12)$$

where $p(k,t)$ is obtained by twice differentiating $\tilde{V}$ with respect to strike price $K$ and applying the change-of-variable $K \to k$:

$$p(k,t) = \frac{\partial^2 \tilde{V}}{\partial K^2}\frac{\partial K}{\partial k} = b(k,t)g(k,t) \quad (13)$$

where the function $g(k,t)$ is given by: $g(k,t) = \left(1 - \frac{k}{\sigma(k,t)}\frac{\partial\sigma(k,t)}{\partial k}\right)^2 - \left(\frac{t\sigma(k,t)}{2}\frac{\partial\sigma(k,t)}{\partial k}\right)^2 + t\sigma(k,t)\frac{\partial^2\sigma(k,t)}{\partial k^2}$. Based on Eq. 10, 11, and 12, we introduce an auxiliary loss function:

$$\min_\theta \frac{1}{|\mathcal{T}|}\sum_{t\in\mathcal{T}}\left(\frac{1}{|\mathcal{K}|}\sum_{k\in\mathcal{K}}\left(\ell_1(k,t) + \ell_2(k,t)\right) + \ell_3(t)\right) \quad (14)$$

Here, $\mathcal{T}$ represents an ordered set of evenly spaced time-to-maturity samples over $[0.01, 2]$ and $\mathcal{K}$ denotes an ordered set of evenly spaced log moneyness samples over $[-1.5, 1.5]$. The notation $|\cdot|$ represents the cardinality of a set. The individual terms of the auxiliary loss function are defined as:

$$\ell_1(k,t) = \max\left(0, -\sigma(k,t) - 2t\frac{\partial\sigma(k,t)}{\partial t}\right) \quad (15)$$

| Index | Interval | Start | End | Interval # | Options # |
|---|---|---|---|---|---|
| SPX | 1-min | 2023-01-03 | 2023-08-31 | 65,151 | 9,606,502 |
| NDX | 1-min | 2023-01-03 | 2023-08-31 | 64,963 | 7,315,928 |
| SPX | 1-day | 2013-01-02 | 2023-08-31 | 2,650 | 17,728,464 |
| NDX | 1-day | 2013-01-02 | 2023-08-31 | 2,681 | 10,432,080 |
| RUT | 1-day | 2013-01-02 | 2023-08-31 | 2,685 | 6,155,780 |
| VIX | 1-day | 2013-01-02 | 2023-08-31 | 1,652 | 388,182 |
| MXWLD | 1-day | 2015-06-08 | 2023-08-31 | 2,023 | 1,524,585 |
| MXEF | 1-day | 2015-06-08 | 2023-08-31 | 2,029 | 1,040,392 |

*Table 2.* Dataset Statistics

$$\ell_2(k,t) = \max(0, -g(k,t)) \quad (16)$$

$$\ell_3(t) = (\mathrm{Trapz}(p(\cdot,t),\mathcal{K}) - 1)^2 \quad (17)$$

where $\mathrm{Trapz}(\cdot)$ denotes the numerical integral using the trapezoidal rule. Empirically, we have found that it is not necessary to tune the weight of Eq.14 when added to Eq.8, as this auxiliary loss will approach zero in the end. Since Eq. 14 depends on a specific reference set $\mathcal{Z}$, we apply it to each reference set within a batch individually.

## 4. Experiments

### 4.1. Dataset

We conduct our experiments using two different time intervals: one-minute and one-day (end-of-day). For the one-minute interval data, we consider index options on S&P 500 (SPX) and NASDAQ 100 (NDX). For the one-day interval data, we additionally include Russell 2000 (RUT), CBOE Volatility Index (VIX), MSCI World Index (MXWLD), and MSCI Emerging Markets Index (MXEF). We filter out in-the-money options, options with prices lower than \$0.1, and options with maturities longer than 2 years. The details regarding data volume after pre-processing can be found in Table 2.

**Train-Test Split** For the one-minute interval data, we train the model using data before 2023-08-01 and test on the subsequent intervals. For the one-day data, we train the model using data before 2023-01-01 and test on the remaining intervals. While each asset-interval has its own dedicated model, totalling 8 models, we also examine the generalization across assets in Sec. 4.5.

### 4.2. Baseline Methods

While our problem setting is new, several existing methods can be adapted to address it. Some deep learning approaches like those of (Ackerer et al., 2020; Zheng et al., 2021) require calibration/training for each time interval from scratch, making them impractical for our scenario due to their parameter complexity, since we have only 9 observed data points. Therefore, we focus on the following adaptable methods:

**SSVI** Hernandez (2016) proposed to map option prices

| Model | SSVI | VAE | GNO | HyperIV |
|---|---|---|---|---|
| Model Param. # | - | 332,289 | 102,529 | 243,153 |
| Imp. Vol. Surf. Param. # | 4 | - | - | 337 |
| Training Time (s) | - | 2.87 | 5.08 | 2.82 |
| Training Memory (MiB) | - | 9,743 | 77,815 | 821 |
| Testing Time (ms) | 13.11 | 0.34 | 7.86 | 2.09 |
| Testing Memory (MiB) | 0.47 | 673 | 2,275 | 532 |

*Table 3.* Model parameters and computational costs

to parameters of an established mathematical finance model (single-factor Hull-White model in their original work) using a neural network. In theory, it can be adapted to predict the parameters of an implied volatility model (e.g., SSVI), however, since calibrating SSVI with 9 option contracts takes comparable time to the forward pass of neural network, we opt to use SSVI directly. Specifically, we employ SSVI's power-law parameterisation to model the total implied variance: $w(k,t|\varsigma,\eta,\gamma,\rho) = \frac{\varsigma^2 t}{2}\left(1 + \rho\eta(\varsigma^2 t)^{-\gamma}k + \sqrt{(\eta(\varsigma^2 t)^{-\gamma}k + \rho)^2 + 1 - \rho^2}\right)$ and the implied volatility is computed as $\sigma(k,t) = \sqrt{\frac{w(k,t)}{t}}$.

**VAE** Bergeron et al. (2021) introduced a method that maps implied volatilities from a fixed grid of deltas and maturities to latent variables. These latent variables are then combined with moneyness-maturity pairs through a decoder to predict the corresponding implied volatilities. This approach can be viewed within the framework of Eq. 6 where $\mathcal{Z}$ is not a set of tuples $\{(k_1,t_1,\sigma_1),(k_2,t_2,\sigma_2),\dots\}$ but rather a vector of volatilities $z = [\sigma_1,\sigma_2,\dots]$, since the coordinates $(k_i,t_i)$ remain consistent across intervals. In our implementation, we upscaled their network architecture to match the parameter count of other baselines, which led to significant performance improvements. Unlike the original study which focused on foreign exchange markets where fixed-grid options are readily available, index options require us to create virtual options since they do not trade on a fixed grid. We provide the details of this virtual grid creation in Appendix B.

**GNO** Gonon et al. (2024) leveraged graph neural operators (Li et al., 2020) to map observed data $\{(k_1,t_1,\sigma_1),(k_2,t_2,\sigma_2),\dots\}$ to a smoothed surface. In their original work, all contracts are treated equally, with the reference set encompassing all observed option contracts in a given interval. To adapt this method to our setting, we limit the reference set to those 9 selected contracts and modify the graph construction as follows: (i) reference set contracts form a fully connected directed graph; (ii) each reference contract has directed edges pointing to all non-reference contracts; (iii) non-reference contracts neither point back to reference contracts nor connect to other non-reference contracts. This graph architecture guarantees that the pre-

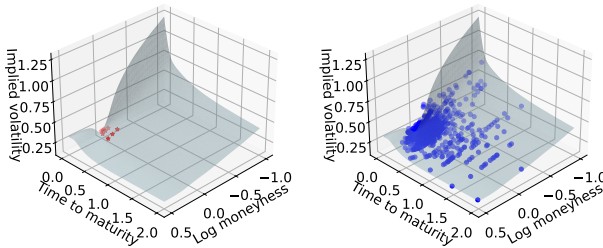

*Figure 1.* Implied volatility smoothing from sparse data. Red stars denote the 9 reference option contracts, blue dots represent test contracts held out from training, and the grey surface illustrates the volatility surface constructed by HyperIV using the reference data.

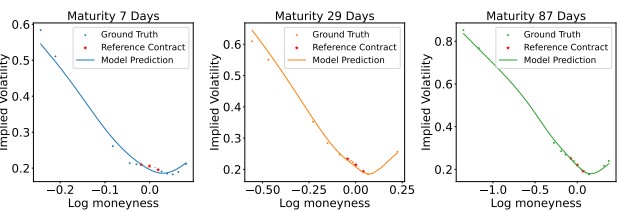

*Figure 2.* The slices of implied volatility surface

diction for any given $(k,t)$ pair depends exclusively on the reference set $\mathcal{Z}$, with complete independence from other queried points on the volatility surface.

The computational characteristics of SSVI, VAE, GNO, and HyperIV are summarised in Table 3. The training time refers to the runtime for one SGD step for a batch of 128 intervals, with each interval containing 16 option contracts. The total training duration spans 500 epochs, with each epoch processing all intervals in batches of 128. The testing time refers to the runtime for constructing the implied volatility surface using a new, unseen reference set and evaluating over a uniform grid of 100 moneyness ($k$) and 100 maturity ($t$) values, totalling 10,000 points. All training and testing procedures are executed on a NVIDIA A100 GPU with 80G VRAM, with the exception of SSVI, which runs on an Intel Xeon CPU.

### 4.3. Qualitative Results

We begin with an illustration of the generated implied volatility surface using SPX end-of-day data on the first testing day (03-Jan-2023). The grey surface in Fig. 1 (left) is constructed using only the 9 reference contracts shown in Table 1. Despite these contracts covering only a small region of log moneyness and maturity, the surface demonstrates excellent fit to the unobserved data, as illustrated in Fig. 1 (right). Further validation is provided in Fig. 2, where we

| Index | Interval | SSVI | VAE | GNO | HyperIV |
|---|---|---|---|---|---|
| SPX | 1-min | 0.0283 | 0.0222 | **0.0140** | 0.0167 |
| NDX | 1-min | 0.0273 | 0.0258 | 0.0162 | **0.0156** |
| SPX | 1-day | 0.0312 | 0.0162 | 0.0085 | **0.0075** |
| NDX | 1-day | 0.0498 | 0.0493 | 0.0117 | **0.0113** |
| RUT | 1-day | 0.0566 | 0.0414 | 0.0133 | **0.0107** |
| VIX | 1-day | 0.2771 | 0.0826 | 0.1345 | **0.0336** |
| MXWLD | 1-day | 0.0511 | 0.0651 | 0.0362 | **0.0166** |
| MXEF | 1-day | 0.0551 | 0.0697 | 0.1040 | **0.0172** |

*Table 4.* Testing MAE (Implied Volatility)

| Index | Interval | SSVI | VAE | GNO | HyperIV |
|---|---|---|---|---|---|
| SPX | 1-min | 3.2698 | 3.6925 | **2.2398** | 2.5115 |
| NDX | 1-min | 15.2799 | 17.3612 | 10.2187 | **10.0043** |
| SPX | 1-day | 6.6257 | 4.0799 | 1.7349 | **1.6736** |
| NDX | 1-day | 25.8445 | 32.2850 | 7.7103 | **7.2159** |
| RUT | 1-day | 2.9377 | 4.5851 | 1.3467 | **0.9554** |
| VIX | 1-day | 0.5256 | 0.1734 | 0.2889 | **0.0771** |
| MXWLD | 1-day | 2.9746 | 8.7424 | 3.2839 | **1.6886** |
| MXEF | 1-day | 3.1641 | 5.7376 | 11.6957 | **1.7596** |

*Table 5.* Testing MAE (Price)

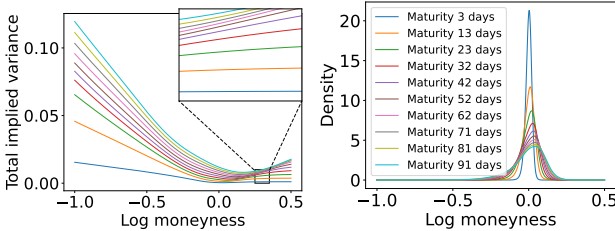

*Figure 3.* Left: the slices of total implied variance $\sigma^2 t$ – the absence of crossed lines suggests no calendar spread arbitrage. Right: the inferred density from implied volatility surface – the absence of negative values suggests no butterfly arbitrage.

| | VAE | GNO | HyperIV |
|---|---|---|---|
| SPX→NDX | 0.0257 | **0.0185** | 0.0223 |
| NDX→SPX | **0.0315** | 0.1272 | 0.0349 |

*Table 6.* Generalization (MAE) on one-minute interval data

| To \ From | SPX | NDX | RUT | VIX | MXWLD | MXEF |
|---|---|---|---|---|---|---|
| SPX | 0.0075 | 0.0154 | 0.0124 | 0.1810 | 0.0159 | 0.0284 |
| NDX | 0.0186 | 0.0113 | 0.0170 | 0.1874 | 0.0202 | 0.0355 |
| RUT | 0.0157 | 0.0134 | 0.0107 | 0.1927 | 0.0164 | 0.0299 |
| VIX | 0.6082 | 0.5606 | 0.2028 | 0.0336 | 0.5377 | 0.2922 |
| MXWLD | 0.0242 | 0.0230 | 0.0232 | 0.1821 | 0.0166 | 0.0203 |
| MXEF | 0.0335 | 0.0267 | 0.0309 | 0.1643 | 0.0221 | 0.0172 |

*Table 7.* Generalization (MAE) on one-day interval data of HyperIV

plot several surface slices corresponding to maturities with reference contracts. The volatility smiles are accurately recovered even without observations for tails, confirming that the hypernetwork has successfully learned the stylised facts.

To verify the absence of static arbitrage, we examine two key aspects. First, Fig. 3 (left) displays the total implied volatilities ($w = \sigma^2 t$) across multiple maturities. According to Eq. 10, total implied volatility should increase monotonically with maturity to prevent calendar spread arbitrage, which is confirmed in Fig. 3. Second, we analyse the inferred density (Eq. 13) for the same set of maturities in Fig. 3 (right). The non-negative nature of these densities rules out butterfly arbitrage.

### 4.4. Quantitative Results

To evaluate the performance of HyperIV and other baselines, we compute the test mean absolute error (MAE) for both implied volatilities and option prices, with results summarised in Table 4 and Table 5. HyperIV achieves superior performance across all assets except for minute-level SPX data, where GNO performs better. Notably, HyperIV demonstrates remarkable stability across all datasets, while SSVI fails on VIX and GNO fails on both VIX and MXEF. It is worth highlighting that SSVI, despite having only four parameters, outperforms VAE (300K parameters) on MXWLD and MXEF.

To investigate the sources of errors, we partition each dataset into 160 two-dimensional bins (10 maturity intervals × 16 log moneyness intervals), ensuring approximately equal numbers of contracts per bin. Fig. 4 displays the mean implied volatilities and mean absolute errors from different models across these bins. Fig. 4 reveals a clear correlation between error magnitude and implied volatility value, with larger errors concentrated in regions of higher implied volatilities (e.g., short maturities and extremely small strikes in the top-left corner). VIX and MXEF exhibit notably different implied volatility distributions compared to other assets, with significant values in the top-right corner (short maturities and extremely large strikes), explaining the poor performance of baseline models on these two assets. In contrast, HyperIV maintains strong performance across all regions. Additional analysis of aggregated errors over time, log moneyness, and maturity intervals is provided in Appendix C, confirming these findings.

### 4.5. Further Analysis

**Generalization** We investigate the cross-asset generalization by examining whether a model trained on one asset can effectively estimate implied volatilities for another asset. Results are summarised in Table 6 and Table 7, with additional baseline comparisons provided in Appendix D. For minute-level data, GNO exhibits strong SPX→NDX gener-

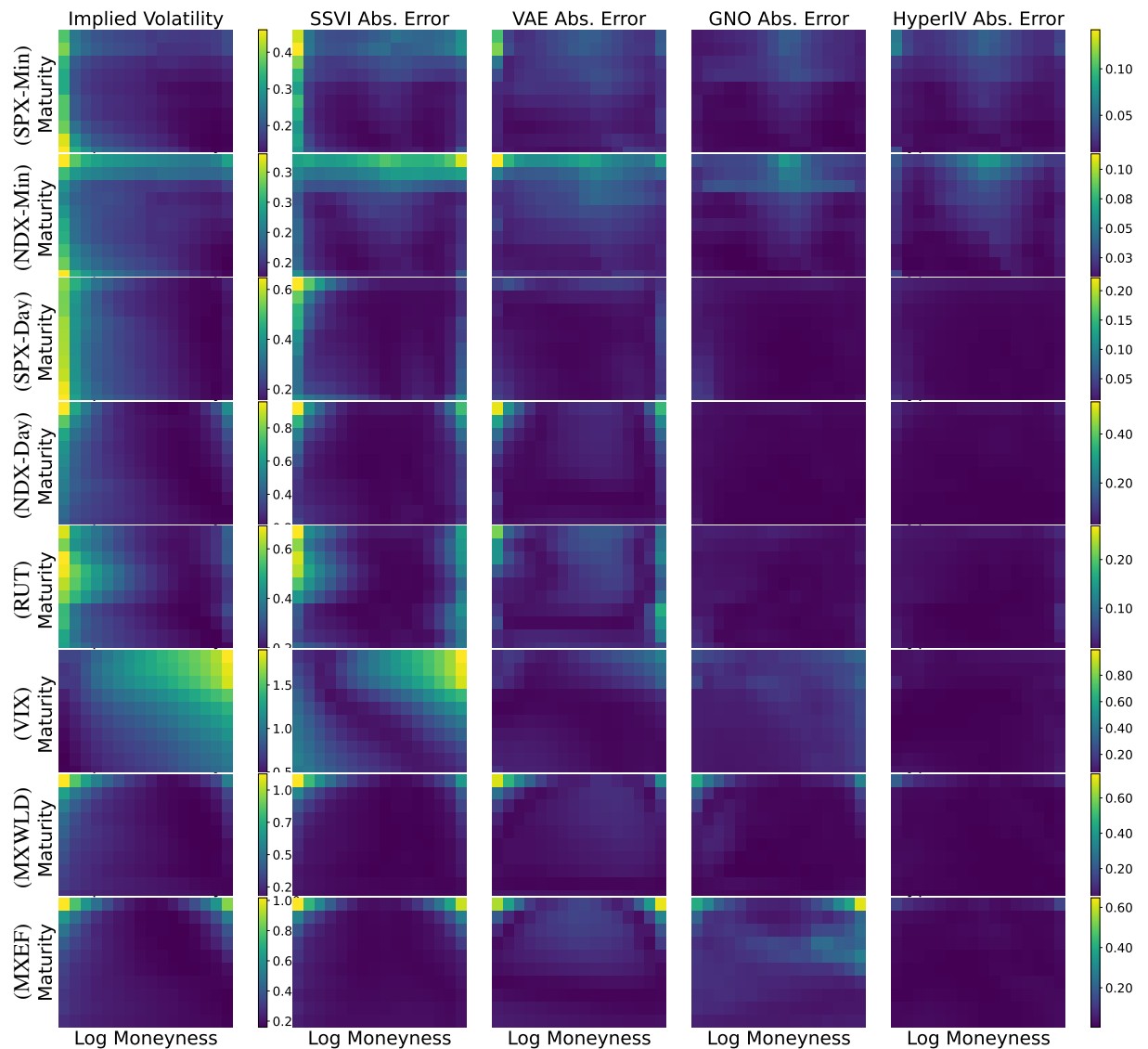

*Figure 4.* Mean values grouped by log moneyness and maturity. The log moneyness is increasing from left to right and the maturity is increasing from top to bottom. The assets are (top-down) SPX, NDX (1-min), SPX, NDX, RUT, VIX, MXWLD, MXEF (1-day).

alization but fails catastrophically in the reverse direction (NDX→SPX). Both VAE and HyperIV demonstrate robust bidirectional performance across directions. For end-of-day data, HyperIV exhibits robust generalization across most assets, with the notable exception of VIX – an expected limitation given the significant domain gap. Remarkably, excluding VIX, HyperIV's generalization performance in any-to-any transfer scenarios surpasses that of SSVI calibrated directly on the target datasets.

**Auxiliary Loss** While our approach does not provide hard guarantees for arbitrage-free conditions, the auxiliary losses remain notably small: approximately $10^{-8}$ during training and $10^{-6}$ during testing. The calendar spread arbitrage (Eq. 15) and positive density function (Eq. 16) constraints

are strictly enforced, as demonstrated by these near-zero loss values. The primary residual error arises from the integrate-to-one constraint (Eq. 17), reflecting minor numerical inaccuracies inherent to trapezoidal integration. In practice, market realities like transaction costs render any residual arbitrage opportunities economically unexploitable, with additional safeguards from standard risk management protocols further mitigating operational risks. This approach of using an auxiliary loss to promote arbitrage-free surfaces is common in contemporary deep learning models (Ackerer et al., 2020; Zheng et al., 2021; Gonon et al., 2024), unlike simpler parametric models such as SSVI, where arbitrage-free conditions are typically guaranteed by analytically constraining the model's parameter space.

## 4.6. Limitation

**Interpretability** Unlike SSVI, our method lacks transparency and interpretability. It has a significantly smaller number of parameters (337) in the implied volatility surface network, but these parameters do not carry meaningful interpretations or provide actionable insights for traders, as SSVI's parameters do. This limitation could be partially addressed through certain architecture designs of the implied volatility model, such as a mixture of SVI/SABR models instead of MLP.

**Computational Cost** While SSVI only needs basic CPU, our method requires modern GPU or NPU. When using an M2 chip instead of an A100 GPU, inference time increases from 2 ms to 8 ms for HyperIV, from 8 ms to 300 ms for GNO, and from 0.34 ms to 15 ms for VAE. Interestingly, SSVI performs better on the M2 chip (from 13 ms to 10 ms) due to better single-core performance.

**Data Accessibility** Due to licence restrictions, we cannot redistribute the data used for model training. Academic researchers may access the end-of-day data through their institution's subscription to WRDS (which includes OptionMetrics). Minute-level data is considerably more expensive and presents greater preprocessing challenges. Theoretically, our method can be verified using synthetic data, which can be generated at any desired frequency.

## 5. Conclusion

We present a novel approach to implied volatility smoothing that effectively addresses the challenges of sparse data and real-time computational constraints. The proposed model, HyperIV, demonstrates how hypernetworks can accurately capture complex implied volatility patterns while simultaneously enforcing arbitrage-free conditions and maintaining computational efficiency. Future research directions include explicitly incorporating historical data and modelling temporal dynamics in implied volatility surfaces.

**Acknowledgements** The authors would like to thank Vincent Zhao at Winton for initially highlighting the problem setting of fitting the implied volatility surface using sparse data, and for insightful discussions throughout this research.

## Impact Statement

The democratisation of options trading through platforms like Robinhood, which has dramatically increased retail investor participation, underscores the critical need for efficient volatility surface smoothing tools. HyperIV provides a streamlined solution where practitioners need only execute a single forward pass on available option quotes to generate a complete arbitrage-free implied volatility surface – eliminating traditional optimisation and hyperparameter tuning. While hypernetwork training incurs substantial upfront computational costs (still significantly lower than baseline methods), the elimination of frequent recalibration reduces long-term operational overhead. This efficiency extends to environmental sustainability, as reduced computational demands translate into lower energy consumption for real-time trading systems.

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

# A. Architecture

## A.1. Neural Network for Implied Volatility Surface

Our model for the implied volatility surface maps coordinates $(k, t)$ to volatility $\sigma$ through three fully connected layers. The first two layers employ hyperbolic tangent activation functions, while the final layer uses a softplus activation function to ensure positive outputs. This architecture contains 337 parameters in total. The PyTorch implementation is shown below:

```
iv_network = torch.nn.Sequential(
    torch.nn.Linear(2, 16),  torch.nn.Tanh(),
    torch.nn.Linear(16, 16), torch.nn.Tanh(),
    torch.nn.Linear(16, 1),  torch.nn.Softplus()
)
```

## A.2. Hypernetwork

The hypernetwork functions as a set embedding network that transforms data from dimension $[B, M, D]$ to $[B, P]$. Here, $B$ represents the batch size, $M$ denotes the number of elements in each set (set to 9 in our implementation), $D$ indicates the number of features per instance (3 features: $k$, $t$, and $\sigma$ for each option contract in the reference set), and $P$ corresponds to the number of parameters in our implied volatility surface network (337 as specified above).

The network architecture consists of several sequential transformations. Initially, a fully-connected layer maps each option's features to $H$ hidden neurons ($[B, M, D] \rightarrow [B, M, H]$). These representations then pass through multiple transformer encoder layers, enabling cross-communication between set elements while maintaining dimensionality ($[B, M, H] \rightarrow [B, M, H]$). The transformer's output is then averaged across the set dimension ($[B, M, H] \rightarrow [B, H]$), and a final fully-connected layer produces the desired parameter vector ($[B, H] \rightarrow [B, P]$). Notably, the absence of positional embeddings in the transformer encoder ensures the network's output remains invariant to the ordering of set elements. The PyTorch implementation follows below.

```python
class SetEmbeddingNetwork(nn.Module):
    def __init__(self, input_dim, output_dim, num_heads=2, num_layers=2,
    ↪   hidden_dim=128):
        super(SetEmbeddingNetwork, self).__init__()
        self.fc1 = nn.Linear(input_dim, hidden_dim)
        self.attention_layers = nn.ModuleList(
            [nn.TransformerEncoderLayer(
                d_model=hidden_dim,
                nhead=num_heads,
                dim_feedforward=hidden_dim,
                batch_first=True,
                dropout=0,
                activation="relu")
            for _ in range(num_layers)])
        self.fc2 = nn.Linear(hidden_dim, output_dim)

    def forward(self, x):
        x = self.fc1(x)
        for layer in self.attention_layers:
            x = layer(x)
        x = x.mean(dim=1)
        x = self.fc2(x)
        return x
```

| | $F$ | $t$ | $r$ | $K$ | $V$ | $k$ | $\sigma$ | $\Delta$ |
|---|---|---|---|---|---|---|---|---|
| 1 | 3827.0768 | 0.0192 | 0.0399 | 3755 | 17.0000 | -0.0190 | 0.2100 | -0.2519 |
| 2 | 3827.0768 | 0.0192 | 0.0399 | 3825 | 42.4000 | -0.0005 | 0.2056 | -0.4867 |
| 3 | 3827.0768 | 0.0192 | 0.0399 | 3900 | 15.0000 | 0.0189 | 0.1955 | 0.2471 |
| 4 | 3836.3443 | 0.0822 | 0.0411 | 3675 | 39.4500 | -0.0430 | 0.2335 | -0.2498 |
| 5 | 3836.3443 | 0.0822 | 0.0411 | 3840 | 92.1500 | 0.0010 | 0.2148 | 0.5061 |
| 6 | 3836.3443 | 0.0822 | 0.0411 | 4000 | 28.3000 | 0.0418 | 0.1936 | 0.2343 |
| 7 | 3853.7277 | 0.2384 | 0.0439 | 3575 | 74.4000 | -0.0751 | 0.2512 | -0.2503 |
| 8 | 3853.7277 | 0.2384 | 0.0439 | 3855 | 163.4000 | 0.0003 | 0.2209 | 0.5203 |
| 9 | 3853.7277 | 0.2384 | 0.0439 | 4125 | 50.4000 | 0.0680 | 0.1917 | 0.2483 |

*Table 8.* A snapshot of the reference set $\mathcal{Z}$ for SSVI, GNO, and HyperIV

| | $F$ | $t$ | $r$ | $K$ | $V$ | $k$ | $\sigma$ | $\Delta$ |
|---|---|---|---|---|---|---|---|---|
| 1 | 3827.0768 | 0.0192 | 0.0399 | 3746.5496 | 18.7273 | -0.0213 | 0.2333 | -0.2500 |
| 2 | 3827.0768 | 0.0192 | 0.0399 | 3827.0768 | 43.7309 | 0.0000 | 0.2070 | 0.5057 |
| 3 | 3827.0768 | 0.0192 | 0.0399 | 3896.9478 | 14.8118 | 0.0181 | 0.1900 | 0.2500 |
| 4 | 3836.3443 | 0.0822 | 0.0411 | 3669.8340 | 40.9677 | -0.0444 | 0.2419 | -0.2500 |
| 5 | 3836.3443 | 0.0822 | 0.0411 | 3836.3443 | 90.5037 | 0.0000 | 0.2070 | 0.5118 |
| 6 | 3836.3443 | 0.0822 | 0.0411 | 3983.0089 | 29.7121 | 0.0375 | 0.1866 | 0.2500 |
| 7 | 3854.9641 | 0.2466 | 0.0440 | 3573.3798 | 75.0444 | -0.0758 | 0.2494 | -0.2500 |
| 8 | 3854.9641 | 0.2466 | 0.0440 | 3854.9641 | 156.2998 | 0.0000 | 0.2070 | 0.5205 |
| 9 | 3854.9641 | 0.2466 | 0.0440 | 4117.5128 | 49.7415 | 0.0659 | 0.1842 | 0.2500 |

*Table 9.* A snapshot of virtual options for VAE

## B. Virtual Options for VAE

SSVI, GNO, and HyperIV can handle continuous data inputs, only requiring the selection of the closest available option contracts from the market. In our implementation, while we target 90-day maturity for options indexed (7,8,9), the actual contracts have 87-day maturity. Similarly, ATM options have log-moneyness ($k$) values close to, but not exactly, zero. For the 25-delta options, we select contracts with delta values nearest to -0.25 for puts and 0.25 for calls. All options presented in the table represent actual market contracts. The reference set $Z$ for the specific example in Table 8 is:

$$\mathcal{Z} = \{(-0.0190, 0.0192, 0.2100), (-0.0005, 0.0192, 0.2056), (0.0189, 0.0192, 0.1955),$$
$$(-0.0430, 0.0822, 0.2335), \quad (0.0010, 0.0822, 0.2148), (0.0418, 0.0822, 0.1936),$$
$$(-0.0751, 0.2384, 0.2512), \quad (0.0003, 0.2384, 0.2209), (0.0680, 0.2384, 0.1917)\}$$

While the ordering of triplets $(k, t, \sigma)$ within $\mathcal{Z}$ is flexible, the component order within each triplet must be maintained.

In contrast to GNO and HyperIV, VAE requires fixed-grid data inputs. To accommodate this requirement, we employ arbitrage-free interpolation (Cohort et al., 2019) to generate virtual options with precise $k = 0$ for ATM options at desired maturities. The interpolation process determines both implied volatilities and corresponding prices, as shown for indices 2, 5, and 8 in Table 9. Subsequently, we optimise $k$ values to achieve exact delta targets: -0.25 for put options (indices 1, 4, 7) and 0.25 for call options (indices 3, 6, 9). For this particular example, the VAE input takes the form of an ordered 9-element vector:

$$z = [0.2333, 0.2070, 0.1900, 0.2419, 0.2070, 0.1866, 0.2494, 0.2070, 0.1842] \tag{18}$$

Unlike the previous case, the order of elements in this vector must strictly follow a specific $(\Delta, t)$ grid pattern: starting with 7-day options (25 $\Delta$ Put, ATM, 25 $\Delta$ Call), followed by 30-day options in the same delta sequence, and concluding with 90-day options in the same pattern.

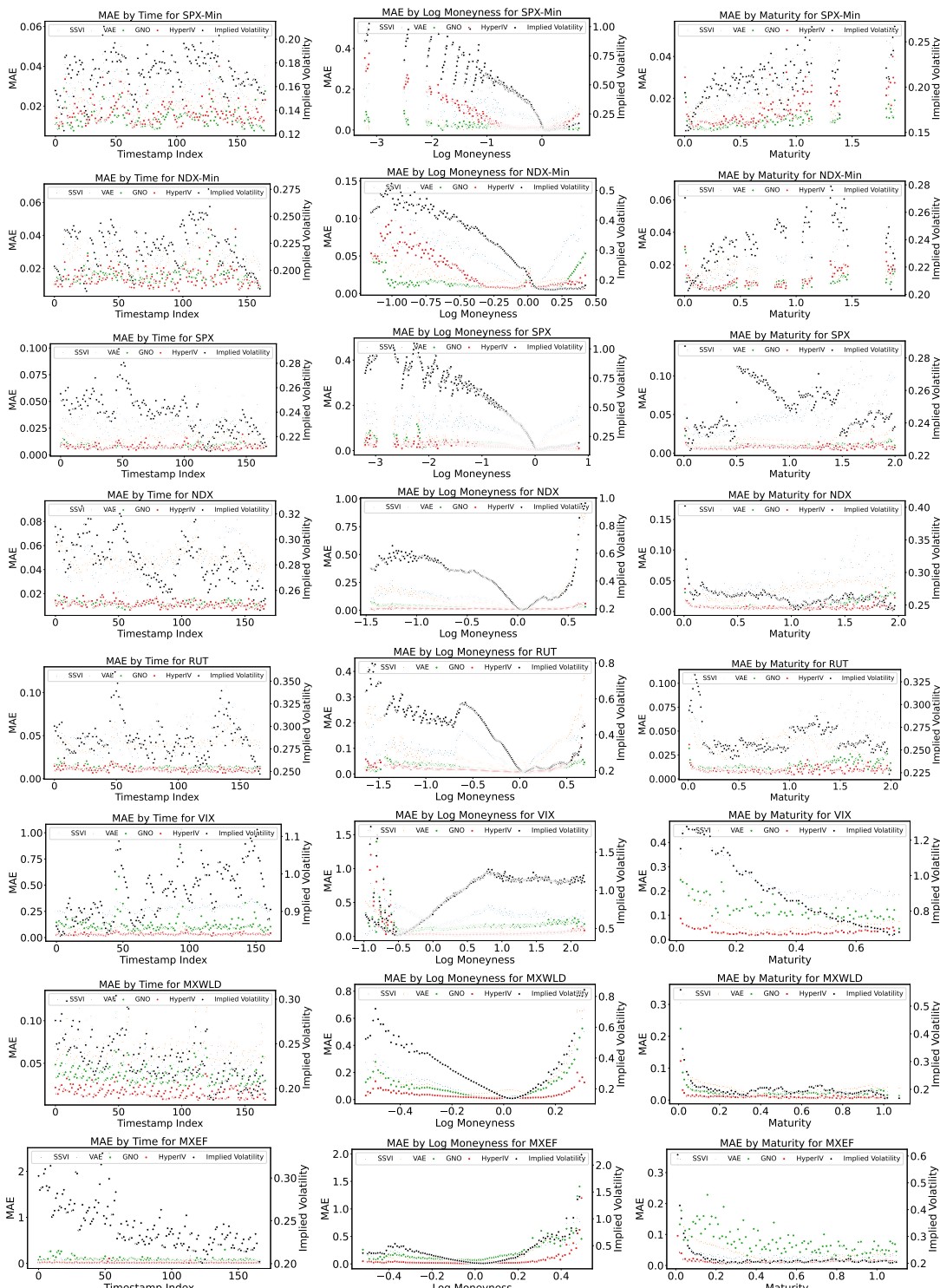

*Figure 5.* Absolute errors aggregated by Time / Log Moneyness / Maturity. The Y-axis on the right is for implied volatility.

## C. MAE by Time, Log Moneyness, Maturity

To analyse the sources of error, we aggregate the absolute errors across different intervals of time, log moneyness, and maturity in Fig. 5. The first column provides insight into model stability over time. HyperIV demonstrates the most

consistent performance across all assets. GNO also performs well generally, with exceptions in VIX and a single outlier in MXEF. While SSVI and VAE show stable performance, their accuracy falls short of both GNO and HyperIV.

The second column illustrates model performance across different ranges of log moneyness (strike prices). The results reveal a clear positive correlation between errors and implied volatility levels. All models struggle with extreme values of log moneyness, with SSVI and VAE showing particular sensitivity to these cases.

The third column presents model performance across different maturity ranges. While less pronounced than the moneyness effect, there remains a notable relationship between implied volatility and errors. All models face challenges in fitting very short maturities, consistent with our findings in Fig. 4 of the main paper. HyperIV maintains consistent performance across all maturities except very short ones, though minute-level data exhibits inherently higher noise levels.

| From
To | SPX | NDX | RUT | VIX | MXWLD | MXEF |
|---|---|---|---|---|---|---|
| SPX | 0.0162 | 0.0444 | 0.0454 | 0.2600 | 0.0508 | 0.0636 |
| NDX | 0.0392 | 0.0493 | 0.0526 | 0.2461 | 0.0584 | 0.0650 |
| RUT | 0.0308 | 0.0422 | 0.0414 | 0.2435 | 0.0481 | 0.0551 |
| VIX | 0.6901 | 0.5960 | 0.6433 | 0.0826 | 0.6170 | 0.5640 |
| MXWLD | 0.0355 | 0.0573 | 0.0631 | 0.2702 | 0.0651 | 0.0783 |
| MXEF | 0.0502 | 0.0570 | 0.0607 | 0.2242 | 0.0661 | 0.0697 |

*Table 10.* Generalization (MAE) on one-day interval data of VAE

| From
To | SPX | NDX | RUT | VIX | MXWLD | MXEF |
|---|---|---|---|---|---|---|
| SPX | 0.0085 | 0.0133 | 0.0146 | 0.0471 | 0.0291 | 0.0797 |
| NDX | 0.0136 | 0.0117 | 0.0175 | 0.0568 | 0.0468 | 0.0812 |
| RUT | 0.0142 | 0.0127 | 0.0133 | 0.0587 | 0.0440 | 0.0765 |
| VIX | 65.0678 | 322.2718 | 305.4467 | 0.1345 | 885.5610 | 200.7394 |
| MXWLD | 0.0196 | 0.0196 | 0.0192 | 0.0508 | 0.0362 | 0.0621 |
| MXEF | 0.0285 | 0.0279 | 0.0319 | 0.0529 | 0.1592 | 0.1040 |

*Table 11.* Generalization (MAE) on one-day interval data of GNO

## D. Further Results on Generalization

We extend our analysis by examining the generalization capabilities of VAE and GNO across different assets. The results reveal that models trained on standard assets fail to generalize to VIX, and conversely, models trained on VIX perform poorly on other assets. This limitation is expected given the distinct implied volatility distribution of VIX compared to other assets.

Interestingly, we observe that models trained on larger datasets sometimes outperform those trained specifically on smaller datasets (such as MXWLD and MXEF). This phenomenon suggests that VAE and GNO benefit significantly from increased training data. In contrast, HyperIV does not exhibit such data dependency, maintaining consistent performance regardless of training dataset size. This distinction highlights a key advantage of HyperIV: its effectiveness in less liquid markets where extensive training data may not be available.

