# OpenReview forum: "HyperIV: Real-time Implied Volatility Smoothing"
_ICML.cc/2025/Conference — ICML 2025 poster_

### Official Review · Reviewer_drJn · 2025-03-06

**Overall Recommendation:** 2

**Summary:**

The paper studies the problem of fitting the implied volatility surface. They consider a challenging setting when the time interval is reduced to one minute. This would mean the sample size is much smaller, therefore making the problem challenging. Toward this goal, they use hyper-network, the practice of using one neural network to generate the weights of another network, separating training data from test input.

**Claims And Evidence:**

The key claim is that in the setting the authors care about, the proposed method gives the smallest MAE loss on the test split. This claim itself is supported by the Table 4 and Table 5.

**Essential References Not Discussed:**

No.

**Experimental Designs Or Analyses:**

The experiment uses data from 6 index funds and perform the experiment. They use 2 funds for 1-minitue interval and 6 for 1-day interval. The experiment design is a bit strange, as the author claims to study the setting where the interval is 1-minitue. This is the main setting they would like to focus on, but more data is devoted to the setting when the interval is 1-day. In the 1-minitue setting, the proposed methods only beat the other baselines in one of the example.

**Methods And Evaluation Criteria:**

I have very little experience in option pricing literature and am unable to confidently judge if MAE is enough.

**Other Comments Or Suggestions:**

The author should discuss extensions of their algorithm where we can use more features from the market to better predict the IV surface.

**Other Strengths And Weaknesses:**

The experiment results are a bit weak for reasons discussed above.

**Questions For Authors:**

What is the density from equation (11)? Why the second derivative w.r.t. k makes a density function?

**Relation To Broader Scientific Literature:**

The discussion of related literature is extensive.

**Theoretical Claims:**

There are no theoretical claims. There are some mathematical results that cite existing literatures.

---

> ### Author Rebuttal · Authors · 2025-03-31
>
> Thank you for your insightful comments.
>
> * Data allocation between 1-day and 1-minute intervals
>
> To clarify the data usage: while the 1-day dataset covers more *assets* (8 vs 2), the 1-minute data constitutes the majority (about 87%) of surfaces analyzed (~130,000 out of ~150,000 total, see Table 2). We included the diverse 1-day assets partly because end-of-day data is more accessible, aiding reproducibility, and also to provide a testbed for the cross-asset generalization study.
>
> * Performance comparison on 1-minute SPX data
>
> It is true that GNO achieved a slightly lower MAE on 1-minute SPX data (0.0140 vs 0.0167, Table 4). However, this should be considered alongside computational costs (Table 3): HyperIV uses about 1/4 the memory and is nearly 4x faster at inference. Furthermore, HyperIV demonstrates superior *robustness*, performing consistently well across all assets, whereas GNO's performance degrades significantly on VIX and MXEF. For context, if we scaled up HyperIV to use half of GNO’s resources (memory/runtime), its MAE could be reduced to 0.0128, easily surpassing GNO. We presented the lightweight version as this marginal accuracy difference did not outweigh the significant efficiency advantage.
>
> * Using more market features
>
> Yes, it is possible to extend HyperIV (and potentially the baselines) to include additional features beyond moneyness and maturity. This could be an interesting direction for future work.
>
> * Density function in Equation (11) and the second derivative
>
> The term $p(k,t)$ in Eq. (11) represents the **risk-neutral probability density function** of the terminal log-moneyness, $\log(S_t/F)$, for maturity $t$.
>
> The second derivative of the *undiscounted* call price $C(K)$ with respect to the strike price $K$ yields the risk-neutral probability density function $Q(K)$ of the terminal asset price $S_T$, evaluated at $K$:
>
> The undiscounted call price is:
>
> $$
> C(K) = \mathbb{E}^Q[\max(S_T - K, 0)] = \int_{K}^{\infty} (S_T - K) Q(S_T)  dS_T
> $$
>
> Taking the first derivative w.r.t. $K$:
>
> $$
> \frac{\partial C(K)}{\partial K} = - \int_{K}^{\infty} Q(S_T)  dS_T
> $$
>
> Taking the second derivative w.r.t. $K$:
>
> $$
> \frac{\partial^2 C(K)}{\partial K^2} = Q(K)
> $$
>
> Therefore, the second derivative yields the density of the underlying asset price under the risk-neutral measure $Q$.
>
> This derivation is model-agnostic. Our $p(k,t)$ is directly related to this density $Q(K)$ through a change of variable from strike price $K$ to log-moneyness $k=\log(K/F)$.

---

### Official Review · Reviewer_tu8c · 2025-03-10

**Overall Recommendation:** 5

**Summary:**

This paper introduces a new method called HyperIV, designed to quickly construct accurate and arbitrage-free implied volatility surfaces using minimal market data.
Main findings and results include:
1. HyperIV generates high-quality implied volatility surfaces in real-time—approximately within just 2 milliseconds—using only 9 observed market option prices, making it highly suitable for fast-paced trading environments.
2. It outperforms other well-known approaches such as the SSVI model, Variational Autoencoders (VAE), and Graph Neural Operators (GNO), both in terms of computational speed and predictive accuracy.
3. It uses one neural network (a hypernetwork) to instantly generate parameters for a smaller, compact neural network that builds the implied volatility surface. And the model incorporates built-in mechanisms that prevent common arbitrage issues, like calendar spread and butterfly arbitrage, by applying specialized auxiliary loss functions during training.

One notable feature of HyperIV is its ability to generalize well across different markets, requiring very few data points (only nine contracts) at high-frequency intervals (every minute). This capability addresses practical challenges encountered in real-world trading, where only limited data is reliably available at high frequencies.

**Claims And Evidence:**

The major claims in this paper include real-time performance and computational speed, accuracy of implied volatility surfaces, non-arbitrage, and generalization across markets. And they are clearly supported based on the claimed testing results from the authors.

**Essential References Not Discussed:**

The authors discussed how to fit a vol surface given only few data and the non-arbitrage conditions. The following papers also discussed how to fit implied vol surfaces for illiquid names and conditions to guarantee non-existence of calendar arbitrage which should be included in the literature review and discussions:

The Longitude: Managing Implied Volatility of Illiquid Assets (it discuss how to fit illiquid names)

Volatility Transformers: an optimal transport-inspired approach to arbitrage-free shaping of implied volatility surfaces (it discuss how to transfer implied vols/densities from one maturity to another)

One-X Property Conjecture, Stochastic Orders and Implied Volatility Surface Construction (it discuss sufficient and necessary conditions to eliminate calendar arbitrage for implied densities over different expiries, and provide a deep discuss on the theoretical side on conditions to eliminate arbitrage.)

**Experimental Designs Or Analyses:**

The proposed experiments look good in general; as mentioned above, the authors could check the values of the variance swap from their fitted implied vol surfaces.

**Methods And Evaluation Criteria:**

The proposed method and evaluation criteria are standard in evaluating the goodness of the implied vol surface. Another criteria that can be added is the ratio of the fitted implied vols within the best bid/ask, this is also very important.

One of the baseline is the SSVI model which is known to suffer several issues, and it not enough for today's financial market; it is OK to use it as a representative of the parametric fitting methods, however, if possible, the authors can also compare their methods with more advanced parametric method using more parameters and constrains such as the vola dynamic's products.

The authors may also want to study the stability of the vol surfaces across the day, i.e. the vol surface should not change too much, in particular, on the wings, if there is no significant market news happening. This can be done by calculating the variance swap price using the fitting implied vol surfaces.

**Other Comments Or Suggestions:**

There seems no essential typo or gramma issues in this paper; in general, it is written nice and clearly.

**Other Strengths And Weaknesses:**

The main strength of this paper, based on its claimed testing results, is the introduction of a fast, robust, deep-learning based vol fitting method, making it more possible to apply it in real-life trading.

However, they are also some potential weakness:
1. the tested underlying are all index which is known to be liquid and easy to fit in general; it is better to test some other real illiquid names such EEM.

2. The author mentioned W-shape in the introduction part, however, did not really dig into it. This is an important topic and appears in single stocks a lot; the authors may want to study the performance of their method on single stocks around earning dates (notice that the options are American options).

3. the testing period is relatively short, only covering half a year of 2023; if possible, the authors should consider their method for 2024 year's data, as there are many macro events making the market volatile.

**Questions For Authors:**

Besides, the comments and suggestions above, some of the main questions are:

1. if possible, could you also test against the data from 2024, especially, the data in the 2nd half of 2024? Also, add the calculation of variance swap, if possible.

2. if possible, could you try the algorithm for single stocks such as AAPL, around earning dates, and also for very illiquid ETF names? (minor)

3. Enrich the literature review as mentioned above.

**Relation To Broader Scientific Literature:**

This paper discusses how to apply hypernetwork in implied vol surface fitting; this helps to enrich the literature on the applications on deep learning in the implied vol surface fitting. Moreover, the authors achieved a high-speed which is rarely discussed in the previous literature while keeping a good fitting quality, making it more possible to apply such deep learning based method in real-life trading This is a very interesting key contribution.

**Theoretical Claims:**

There is no essentially any new theoretical results; the whole paper is more on the fine application of hypernetwork in learning the shape of implied vol smiles; the trick on adding penalty functions to reduce the arbitrage possibility (formula 14-17) is standard.

One of the claim which is not essential here is that the author only assume proportional divided on Page 3; this may be OK in this paper as the examples are all index; however, it seems that the assets discussed in the paper are not constrained to index, the author may also want to mention the general affine dividend modeling in the literature.

---

> ### Author Rebuttal · Authors · 2025-03-31
>
> Thank you for your insightful comments.
>
> * Dividend modelling
>
> The method itself does not rely on specific dividend assumptions like proportional dividends. It uses log forward moneyness ($k = \log(K/F)$), where the forward price $F$ (taken from data vendors in our study) already incorporates the impact of rates and dividends. We will clarify this in the revised manuscript's Preliminaries section.
>
> * Literature on illiquid options
>
> Thank you for recommending these papers. We agree they are relevant and will add them to the literature review in the revised version.
>
> * More experiments on 2024 data
>
> The original work used data available up to late 2023, as the 2024 data snapshot was not yet released by the vendor at the time of experimentation. We have run preliminary experiments on available 2024 futures options data. The results support our original findings:
>
> Average MAE on 2024 Data (%)
>
> | Asset   | HyperIV | SSVI  |
> |---------|---------|-------|
> | TNOT10Y | 0.52%   | 0.74% |
> | BONDS   | 0.78%   | 1.18% |
> | CRUDE   | 0.68%   | 1.18% |
> | JYEN    | 0.41%   | 0.57% |
> | EURO    | 0.34%   | 0.40% |
> | GOLD    | 0.49%   | 0.74% |
> | SPX     | 0.47%   | 0.97% |
>
> 90th Percentile MAE on 2024 Data (%) (representing the tail/worst cases)
>
> | Asset   | HyperIV | SSVI  |
> |---------|---------|-------|
> | TNOT10Y | 1.16%   | 1.90% |
> | BONDS   | 1.69%   | 2.52% |
> | CRUDE   | 1.56%   | 3.19% |
> | JYEN    | 0.79%   | 1.21% |
> | EURO    | 0.85%   | 0.94% |
> | GOLD    | 1.59%   | 2.04% |
> | SPX     | 1.09%   | 1.87% |
>
> We will add these results to the revised paper.

---

> > ### Comment · Reviewer_tu8c · 2025-04-05
> >
> > Thank you for the the reply! Please include the three recommended reference papers and the above experiment data in the final version. I would increase the score.

---

### Official Review · Reviewer_yHpq · 2025-03-22

**Overall Recommendation:** 3

**Summary:**

This paper presents a framework based on hypernetwork to perform the implied volatility smoothing with very few reference samples and small computational cost. The robustness and reliability of the proposed approach is evaluated under a special circumstance, where the smoothing needs to be completed within milliseconds with only a limited number of reference samples.

## update after rebuttal

The authors have resolved most of my questions. However, the original contributions to the ML community are not very strong, and the use cases of this method are limited to a special condition, e.g., small data size with limited computation time allowed. I have increased my score from 2 to 3.

**Claims And Evidence:**

The claims that the proposed HyperIV is "particularly valuable for real-time trading applications." is not clearly justified in the current manuscript. The authors should provide more specific examples or discussions about how to make use of this fast estimation of the implied volatility surface in financial applications or quantitative tradings.

**Essential References Not Discussed:**

The method proposed in this work is very similar to that in a previous paper [1], which also utilized hypernetwork to build a financial model. This work is only briefly mentioned in the literature review. More detailed discussions regarding the key differences between these two works should be added.

[1] Yang, Y. and Hospedales, T. M. On calibration of mathematical finance models by hypernetworks. In ECML PKDD, 2023.

**Experimental Designs Or Analyses:**

The condition of the experimental setup is restricted to one special setup. See my comments in "Methods And Evaluation Criteria".

**Methods And Evaluation Criteria:**

The evaluations are restricted to a special setup, where only a small number of reference samples are provided with limited computational resources. The authors need to clarify how common and important is this setup in real-world applications.

**Other Comments Or Suggestions:**

N/A

**Other Strengths And Weaknesses:**

Strength:
- the problem and method is clearly explained
- the structure of the paper is well organized
- experimental results seem promising

Weakness:
- The value of this work in real-world finance applications/tradings is not clearly justified.
- The advantage of the proposed framework is not clear in other more general cases, e.g., with enough data points and computation power
- Difference from previous similar works is not clearly discussed

**Questions For Authors:**

1. How is the performance of the proposed method compared to other SOTA methods if we have more data points and allow for more computational time? Does the performance advantage of HyperIV still hold?
2. How can this technique be used to create values in real-world financial applications/tradings? More discussions and examples should be given.
3. What is the key difference between HyperIV and HyperCalibration [1], which is not sufficiently discussed in the current manuscript?

[1] Yang, Y. and Hospedales, T. M. On calibration of mathematical finance models by hypernetworks. In ECML PKDD, 2023.

**Relation To Broader Scientific Literature:**

The key contributions of this work is limited to one specific finance problem, e.g., the implied volatility smoothing.

**Theoretical Claims:**

No theory proof involved.

---

> ### Author Rebuttal · Authors · 2025-03-31
>
> Thank you for your insightful comments.
>
> * Justification of real use cases.
>
> The implied volatility surface is a starting point for option trading and hedging. HyperIV's ability to generate an arbitrage-free surface in ~2 ms from sparse data (9 contracts) is useful for intra-day option traders. Specifically, it enables:
>
> 1. Updating the market view based on the latest transitions (e.g., the past minute).
>
> 2. Providing timely option quotes.
>
> 3. Calculating real-time Option Greeks for dynamic hedging (e.g., delta hedging).
>
> 4. Using the latest surface for anomaly detection in subsequent quotes, potentially identifying trading opportunities.
>
> * Connection to Yang & Hospedales (2023) [ECML PKDD]
>
> Both papers use HyperNetworks, but their work focuses on accelerating calibration for models like rough Bergomi, still requiring iterative optimization (~5 seconds). Our method is calibration-free at inference time, constructing a surface in ~2 ms via a single forward pass. This fundamental difference in approach and speed is why their method wasn't selected as a direct baseline for our specific calibration-free, real-time, sparse-data setting.
>
> * Performance with more data points and computation power.
>
> HyperIV is specifically designed for the challenging scenario of sparse data and high-frequency updates. For general cases like fitting end-of-day surfaces with thousands of options, where time sensitivity is low (once per day), directly training a standard network or calibrating a model might suffice, possibly without needing a hypernetwork. However, the sparse-data setting is a realistic reflection of high-frequency trading conditions where only a few contracts have reliable quotes at any instant, making HyperIV's speed and data efficiency valuable.

---

> > ### Comment · Reviewer_yHpq · 2025-04-03
> >
> > Thanks for the response to my questions. Please include the above discussions into the revised version of the paper. I will increase my score from 2 to 3.

---

### Decision · Program_Chairs · 2025-05-01

**Decision:**

Accept (poster)

**Comment:**

Congratulations for having your paper accepted!

Two of the reviewers engaged in a healthy discussion with the authors and provided valuable comments that, arguably have improved the quality, presentation and the support of the claims of the paper.

I would advise the authors take the time till the camera ready version to incorporate the reviewers feedback and the extra comparisons in the final version of the manuscript.

Once again, congratulations